# Brain Metabolic Profile after Intranasal vs. Intraperitoneal Clomipramine Treatment in Rats with Ultrasound Model of Depression

**DOI:** 10.3390/ijms22179598

**Published:** 2021-09-04

**Authors:** Olga Abramova, Yana Zorkina, Timur Syunyakov, Eugene Zubkov, Valeria Ushakova, Artemiy Silantyev, Kristina Soloveva, Olga Gurina, Alexander Majouga, Anna Morozova, Vladimir Chekhonin

**Affiliations:** 1V. Serbsky National Medical Research Centre of Psychiatry and Narcology, 119034 Moscow, Russia; abramova1128@gmail.com (O.A.); zubkov@ngs.ru (E.Z.); ushakovavm@yandex.ru (V.U.); artsilan@gmail.com (A.S.); olga672@yandex.ru (O.G.); hakurate77@gmail.com (A.M.); chekhoninnew@yandex.ru (V.C.); 2Mental-Health Clinic No. 1 Named after N.A. Alekseev, 117152 Moscow, Russia; sjunja@bk.ru (T.S.); soloveva.kr@yandex.ru (K.S.); 3Federal State Budgetary Institution Research Zakusov Institute of Pharmacology, 125315 Moscow, Russia; 4Drug Delivery Systems Laboratory, D. Mendeleev University of Chemical Technology of Russia, Miusskaya pl. 9, 125047 Moscow, Russia; rector@muctr.ru; 5Department of Medical Nanobiotechnology, Pirogov Russian National Research Medical University, 117997 Moscow, Russia

**Keywords:** metabolome, rats, depressive-like behavior, hippocampus, frontal cortex

## Abstract

Background: Molecular mechanisms of depression remain unclear. The brain metabolome after antidepressant therapy is poorly understood and had not been performed for different routes of drug administration before the present study. Rats were exposed to chronic ultrasound stress and treated with intranasal and intraperitoneal clomipramine. We then analyzed 28 metabolites in the frontal cortex and hippocampus. Methods: Rats’ behavior was identified in such tests: social interaction, sucrose preference, forced swim, and Morris water maze. Metabolic analysis was performed with liquid chromatography. Results: After ultrasound stress pronounced depressive-like behavior, clomipramine had an equally antidepressant effect after intranasal and intraperitoneal administration on behavior. Ultrasound stress contributed to changes of the metabolomic pathways associated with pathophysiology of depression. Clomipramine affected global metabolome in frontal cortex and hippocampus in a different way that depended on the route of administration. Intranasal route was associated with more significant changes of metabolites composition in the frontal cortex compared to the control and ultrasound groups while the intraperitoneal route corresponded with more profound changes in hippocampal metabolome compared to other groups. Since far metabolic processes in the brain can change in many ways depending on different routes of administration, the antidepressant therapy should also be evaluated from this point of view.

## 1. Introduction

Depression is a serious mental disability characterized by low mood accompanied by a disturbed sleep and/or appetite, loss of interest or pleasure, feelings of guilt or low self-worth, and cognitive dysfunctions such as decreased memory, attention, learning, and goal-seeking behavior. Major depressive disorder is common, with almost one in five people experiencing one episode at some point in their lifetime [1]. In addition to cognitive impairment, depression is characterized by problems in social functioning and ability to work. Depression affects an individual’s quality of life, their families, and wider society [2].

Being a widespread severe illness, depression makes scientists seriously consider its pathogenesis in order to develop effective methods to prevent it or cope with consequences of the disease.

Experimental rodent models for the simulation of depression are still among the most promising sources of information about biological mechanisms involved in the pathology of depression. An increasing attention has been focused on the influence of stress over depression. The link between stress and depression has been known for a long time. Hypothalamic–pituitary–adrenal axis hyperactivity is one of the commonest neurobiological changes in depressive patients. Dysfunction of hypothalamic–pituitary–adrenal axis is manifested in about 70% patients with clinically diagnosed depression, as studies over the many past years show [3]. At the same time, an increase in cortisol levels is the primary reaction of the body in response to stress. We suggest that stress-induced models are the most appropriate for these goals, and there are plenty of them: for example, chronic mild stress model [4], repeated restraint stress model [5], learned helplessness model [6], olfactory bulbectomy model [7], and others [8]. However, it is questionable whether the results of these studies can be extrapolated to humans, because the protocols for modeling depression involve mostly physical stressors, not psychological ones, without considering that psychological stress, such as terrorism, war, divorce, and unemployment, are the main stressor factors for humans in the present world [9]. There are some animal models based on psychological impact such as maternal deprivation model [10], social isolation model [11], social defeat stress model [12], but all of them are influenced by one stress factor which is not typical for a human life in modern society. There are multiple intrinsic and extrinsic factors that may trigger the onset of depression. One of the most dramatic is massive information impact, which is characteristic of the modern urban world. It is not only about the excess amount of information which comes from television, internet, and press, but it is also about its inconsistency, which evokes controversial emotions. In this paradigm, it is important to continue studies aiming to make a valid model of depression in order to be able to translate findings to people.

We have created an experimental depression model based on the negative information impact, which is mediated by ultrasound (US) waves [13]. Moreover, we have shown that the model has face validity, when animals recapitulate disease phenotype in a similar way to depressed humans [14], and predictive validity (pharmacologic sensitivity), when animals respond to medicines that are effective to treat depressed humans [13,14]. We show that in our model, serotonin metabolism is impaired and production of brain-derived neurotrophic factor in the hippocampus is reduced, which resembles pathophysiological processes that cause a disease in humans and can be a part of a construct validity (relevance) [13]. Alterations in the hypothalamic–pituitary–adrenal axis are observed in depression and also shown in animal models. In the US model of depression, the level of epinephrine, norepinephrine, and corticosterone was elevated after US stress. The decreased levels of serotonin, dopamine, and their metabolites were observed in blood [14].

A number of studies show that metabolomics profiles are different in depressed individuals compared to healthy controls [15]. For example, higher concentrations of amino-acids, glutamate, aspartate, and cysteine have been observed with the dysfunction of fatty acids in patients with depression [16]. Together with this, changes are found in compositions of lipid metabolites and neurotransmitter metabolites in the blood (dicarboxylic fatty acids, glutamate, aspartate, γ-aminobutyric acid (GABA), citrate, glycerate, 9,12-octadecadienoate, and glycerol) [17]. Downregulated hippuric acid, N-methyl-nicotinamide, and upregulated azelaic acid have been found in the urine of patients with depression [18].

MRI spectroscopy shows a decrease of *N*-acetylaspartate, glutamate, creatine, GABA, and phosphocreatine and an increase of choline and lactate in the brains of individuals with depression [19]. Clearly, disruption of a large number of metabolic pathways is associated with depression (anaerobic glycolysis may explain elevated lactate levels, aspartate is involved in the synthesis of glutamate, and *N*-acetylaspartate is ubiquitous in neurons).

Significant differences in metabolic phenotypes between non-medicated depressed patients and healthy controls were revealed, whereas differences between non-medicated and medicated patients were found to be insignificant [20].

Metabolomic studies show significant perturbations within and between the tryptophan, tyrosine, purine, and kynurenic acid pathways due to antidepressants exposure. Changes in the methoxyindole pathway and kynurenine/tryptophan ratio were correlated with treatment outcomes [21,22].

Despite the large number of studies of the metabolome of depression, rather few of them have investigated metabolic changes in response to antidepressants, which would be interesting in light of the evidence on the low effectiveness of antidepressants [23]. In addition, studies of metabolic disturbances directly in the brain tissue are very difficult to do for humans. Animal studies might offer a valuable option to gain a deeper insight into the mechanisms underlying depression and antidepressants action. The metabolomic analysis of brain tissues helps to identify whether the effects of diseases can be seen throughout the whole body. There are many studies about metabolic abnormalities in animal models of depression [24,25,26].

Maintenance of the metabolic balance is necessary to preserve homeostasis and healthy living. Any exposure causes some metabolic imbalance, and as soon as the adaptation limits are exceeded, there comes a cascade of chemical reactions causing metabolic imbalance manifesting in the disease symptoms. Medicine applied affects metabolic processes too, and the ones that take place in brain tissue are of the most interest.

We aimed at studying the variable frequency US impact on rats’ brain tissue. We then observed how tricyclic antidepressant clomipramine administrated in different ways affects metabolic processes in animal models and their behavior. To achieve these goals, we did the following:We accessed the depressive-like animal state and also their cognitive functions after regular exposure to the variable frequency US waves in groups that got no treatment, and those that got treatment by clomipramine applied intranasally and intraperitoneally.We analyzed the amount and composition of 28 metabolites in rats’ frontal cortex (FC) and hippocampus in a normal and depressed-like state after treatment and with no treatment at all.We studied metabolic pathway changes related to depressive-like state and tried to find the metabolic pathway which could made the greatest contribution to metabolomic changes in depression.We made a comparative analysis of metabolic pathway changes depending on intranasal and intraperitoneal routes of drug administration.

## 2. Results

### 2.1. Behaviour Tests

According to the Shapiro–Wilk test for normality, the data significantly deviated from a normal distribution only in the ultrasound group, which is reflected in time spent to find the platform (escape latency) in the Morris water maze test.

The sucrose preference test determines the degree of anhedonia in animals. Anhedonia is observed in depressive disorders. ANOVA revealed significant effects of the animal group for Anhedonia (F = 7.42, df = 3, *p* = 0.006, η^2^ = 0.36). Further post hoc analysis showed the decreased index in the US group and differences between the US control group (*p* = 0.0004) and the US IN clomipramine group (*p* = 0.019). Therefore, the US group exhibited reduced sucrose preference, while after intranasal administration of clomipramine, the results were close to the control group level and the did not differ from it (Figure 1A).

The social interaction test indicates interest in social contacts in rats, which may be decreased in the depressive-like state. ANOVA revealed significant effects of the animal group for social interaction (F = 4.52, df = 3, *p* = 0.010, η^2^ = 0.31). In the social interaction test, the time spent in social interaction decreased in the US group. After antidepressant treatment, the social interaction time increased to the control group level (US-IN Clom *p* = 0.027, US-IP Clom *p* = 0.009, US-Control *p* = 0.05 (Figure 1B)). Thus, treatment with antidepressants reduced a symptom of depressive-like behavior such as decreased social activity.

A forced swim test was performed to assess antidepressant-like effects of drugs. ANOVA revealed significant effects of the animal group for the forced swim test (F = 48.94, df = 3, *p* < 0.001. η^2^ = 0.79). US group rats showed the longest immobility time (*p* < 0.0001) compared to all the other groups, which did not differ (Figure 1C).

Cognitive impairment is a common symptom in depression, and the effects of clomipramine on some aspects of cognitive performance were also evaluated. The Morris water maze test assesses long-term spatial memory. Our study showed differences between groups (Kruskal–Wallis chi squared = 8.3125, df = 3, *p*-value = 0.04). The rats after US stress showed impairment of this type of memory. The US group rats needed more time to find the platform compared to the control group (*p* = 0.012) (Figure 1D). The rats treated with clomipramine showed no difference from the control and US group.

### 2.2. Metabolomic Analysis in Brain

Metabolites in the hippocampus and FC were determined. The results of the differences between the groups are presented in Table 1.

### 2.3. Principal Component Analysis (PCA)

The PCA biplots are plotted in Figure 2. For both frontal cortex and hippocampus, five principal components were produced which explain 96% and 85% of metabolites’ profile variability, respectively, while the first two components account for 66% of each brain structure metabolite content variability (scree plots in the Supplements Appendix A, Figure 2).

PCA results indicate that metabolic profiles differ between animal groups. Post hoc comparison of regression slopes of principal components 1 and 2 scores in animal groups (Table 2) revealed that the general metabolic profile in the hippocampus in the IP Clom group significantly differ from every other group, while in the frontal cortex, the IN Clom group differs from the control and US groups.

### 2.4. Heatmap Visualization

Heatmap visualization of metabolomics data in the animal groups for hippocampus and frontal cortex can be seen on the Figure 3. Heatmaps of the metabolic profiles for individual animals and heatmaps of group mean raw concentrations with pairwise comparison between groups are provided in the Supplement (Appendix A). Heatmaps visualize distinct patterns of metabolites content distribution in the animal groups.

### 2.5. Metabolic Pathways Analysis

To explore the possible pathways affected by US stress and clomipramine effects, the 28 metabolites were imported into the Pathway Analysis module in MetaboAnalyst 5.0. The results presented in Figure 4 and details of pathways were provided in Appendix A. We prepared metabolic pathways analysis in the frontal cortex between the control group and the other groups. The nine metabolic pathways, i.e., (1) tryptophan metabolism, (2) aminoacyl-tRNA biosynthesis, (3) alanine, aspartate, and glutamate metabolism, (4) arginine and proline metabolism, (5) arginine biosynthesis, (6) d-Glutamine and d-glutamate metabolism, (7) pentose phosphate pathway, (8) nicotinate and nicotinamide metabolism, and (9) glycine, serine, and threonine metabolism, are recognized as the most influenced metabolic pathways associated with US stress in frontal cortex (Appendix A, Figure 4). One pathway, i.e., purine metabolism, is found to be involved in the antidepressant-like effects of peritoneal clomipramine on US rats (Appendix A, Figure 4). A pathway analysis provides information about biological roles of metabolites, which can further provide insights into pathogenesis and mechanisms of depression or antidepressants’ effects.

## 3. Discussion

In our study, we investigated the effect of variable frequency US on the behavior of male rats, taking into account the antidepressant clomipramine therapy with different routes of administration. We studied changes in the metabolomics profile of rat brain regions under the influence of these factors.

Animals exposed to US show evident depression-like states including reduced preference for sucrose, less social interaction time, increased immobility time in the forced swim test, and low Morris water maze cognitive performance: 50% of rats failed to find the platform. These findings correlate with the data we obtained from our previous studies of depression-like states in rats induced by variable frequency US exposure.

Clomipramine treatment reduced immobility time compared to the control group. Treated group has the same index of sucrose preference as the control. US group has the lowest index Clomipramine treatment the time to find the platform; but there were no dif-ference between both US and control group. Clomipramine treatment did not affect social interaction time, which was similar to the group with no treatment. The obtained data correlated with the results of other scientists’ research. Clomipramine has moderate antidepressant properties [23]. There was no difference found between intranasal and intraperitoneal modes of administration concerning drug efficacy in our study.

The other aim of our study was to examine the metabolomic profile of rats’ frontal lobe and hippocampus in the four groups. Chronic stress plays a crucial role in the etiopathology of psychiatric disorders such as anxiety and depression. Prefrontal lobe dysfunction is pathophysiologically linked to stress-related cognitive and emotional disorders [27]. The hippocampus is a part of the brain that plays a key role in pathogenesis of depression and memory disorders [25]. Moreover, it is one of the adult neurogenic regions of the brain. Disrupted neural plasticity induces depressive disorders, and methods that boost neurogenesis have a therapeutic effect [28].

The most significant changes of the metabolomics profile were observed in the cerebral cortex. The metabolomics profile of the control group differed as compared to other groups. The group who had an antidepressant treatment was akin to the group who had no treatment at all. According to metabolic pathways analysis, changes in the brain cortex affected the following pathways: tryptophan metabolism, aminoacyl-tRNA biosynthesis, alanine, aspartate, and glutamate metabolism, arginine and proline metabolism, arginine biosynthesis, d-Glutamine and d-glutamate metabolism, pentose phosphate pathway, nicotinate and nicotinamide metabolism, glycine, and serine and threonine metabolism.

The pathways analysis demonstrated changes in an important metabolic pathway for depression, the tryptophan metabolism pathway. The cortical tryptophan concentration decreased in all three groups exposed to US stress. At the same time, clomipramine had no effect on tryptophan metabolism. Tryptophan is metabolized by two main pathways: the kynurenine pathway and the serotonin pathway; most of it is converted to kynurenine. It was shown that depressive disorders, including those induced on animal models, activated the kynurenine pathway and its toxic metabolites may be involved in causing or triggering depression [29]. Tryptophan metabolism is linked to depressive disorders and its modulation could be linked to some of the antidepressant efficacy, which was demonstrated in some rat models [30]. Therefore, our study showed that the effect of chronic US variable frequencies alters the metabolic pathway that is associated with depression, which additionally indicates a depression-like state in animals.

The glutamate neurotransmitter system is also associated with depression. Our study demonstrated altered alanine, aspartate, and glutamate metabolism and d-glutamine and d-glutamate metabolism pathways of the rat frontal cortex under the influence of variable US. Glutamate is the main excitatory neurotransmitter in the brain. Additionally, the glutamatergic system can modulate the activity of other neurotransmitter systems [31]. The glutamate–glutamine cycle is a major glutamate metabolic pathway and it is crucial for the normal glutamatergic neurotransmission. The biological sense of the glutamate–glutamine cycle is the following: glutamate needs to be removed from the synaptic cleft; glutamate is converted to glutamine which is its inert form, so that it can be transported back to the neuron cell body. Finally, ammonia, which is known to be a potent neurotoxin, is disposed by astrocytes as far as the urea cycle takes place only in the liver. The glutamatergic system dysregulation plays a role in depression pathophysiology [25]. Zink et al. [32] showed a crucial role of glutamate in the development of a depression-like state induced by learned helplessness. Of all the depression models, this one most closely resembles US depression because it implies no physical impact, just psychological stress that the animal experiences. Glutamate–glutamine cycle dysfunction in rats’ blood and brain tissue was shown in the chronic mild stress model [33], which manifested in low glutamine and GABA concentration in rats’ brain tissue. Similar observations were made in our survey. Meta-analysis with 1180 patients and 1066 healthy controls demonstrated a significant decrease in the glutamine level of the medial frontal cortex [34]. It was shown that exogenous glutamine administration could have antidepressant effects by promoting glutamatergic neurotransmission and reduce the depression-like behavior in mice exposed to chronic immobilization stress [35]. The therapeutic response to cytidine in bipolar depression patients was explained by its ability to decrease glutamine/glutamate concentration in the brain tissue [36].

In our study, there was an increased concentration of aspartic and glutamic acids in rats’ prefrontal cortex. As it was said before, aspartic acid also plays an essential role into glutamate glutamine cycle. Aspartic acid is also a part of glutamate receptors. There is now a wide range of effective antidepressants targeting *N*-methyl-d-aspartate receptors, which proves the effectiveness of different glutamatergic modulators including (1) broad glutamatergic modulators (ketamine, esketamine, dextromethorphan, a combination of dextromethorphan and quinidine (Nuedexta), AVP-786, nitrogen oxide [N_2_O], AZD6765), (2) subunit (NR2B)-specific *N*-methyl-d-aspartate receptor antagonists (CP-101,606/traxoprodil, MK-0657 [CERC-301]), (3) glycine site modulators (d-cycloserine, GLYX-13, sarcosine, AV-101), and (4) metabotropic glutamate receptor modulators (AZD2066, basimglurant, AZD2066, RO4917523/basimglurant, JNJ40411813/ADX71149, R04995819 [RG1578]) [37]. Alanine, aspartate, and glutamate pathways contribute to the development of depression, which was proved by metabolomic analysis of rats exposed to prenatal stress [38]. Assuming that alanine, glutamate, and glutamine are involved in energy exchange, in the citric acid cycle in particular, chronic exogenous stress could lead to energy exchange disruption in rats’ brain tissue [39]. Metabolomic analysis of mononuclear cells from peripheral blood of rats exposed to chronic stress demonstrated defects of neurotransmitter metabolism, energy metabolism, and oxidative stress metabolism, such as lower concentration levels of aspartic and glutamic acids as compared to healthy rats [40].

In our study, there was a decrease in GABA concentration in the PFC and hippocampus in the US group. Meanwhile, intraperitoneal administration of clomipramine resulted in significantly increased hippocampal GABA levels compared to control and US groups, whereas the effectiveness of clomipramine administered intranasally was comparable to the control group. Nevertheless, cortical GABA levels decreased in both cases compared to control and US groups (Table 1, Supplements Appendix A). Patients diagnosed with major depressive disorder or bipolar disorder are known to have impaired excitatory and/or inhibitory neurotransmission and neuroplasticity. Neural network dysfunction resulting from glutamate and GABA concentration changes was detected when studying both animal and human depression. The connection between these systems concerning the role they play in depression pathogenesis is proved by ketamine’s antidepressant effect [41]. The neurobiological basis of these changes was studied concerning both glutamate excitatory neurons and GABA inhibitory interneurons. Structural, functional, and neurochemical disorders were observed, which could result in corrupted signal integrity in the brain cortex and hippocampus. The molecular basis of those changes is not quite clear, but it is thought that there is a connection to stress-induced excitotoxicity, increased secretion of glucocorticoids from the adrenal gland and inflammatory cytokines, and also the interference of some environmental factors [42]. GABA is a major inhibitory neurotransmitter in the brain. GABAergic neurons account for 20–40% of all neurons in the cerebral cortex depending on its region and they tune up excitatory neurotransmissions of different neuronal systems [43]. Chronic stress and depression are linked to GABAergic dysfunction, including atrophy of pyramidal neurons and a decrease in GABAergic markers in the medial PFC and hippocampus. Rapid-acting antidepressants, especially ketamine and esketamine, which are characterized by rapid onset of antidepressant action, are thought to activate GABA receptors in the brain [42]. The antidepressants which are used to enhance monoaminergic transmission are all characterized by their ability to increase GABAergic transmission [43,44].

According to the results of frontal cortex metabolic pathways analysis in our study of US-induced depressive-like states, the cerebral pentose phosphate pathway appeared to be of great importance. Additionally, we found decreasing concentration of ribose-5-phosphate, which is essential for nucleotide biosynthesis, in the frontal cortex of the rats exposed to US. The pentose phosphate pathway is an alternative way of oxidizing glucose. The pentose phosphate pathway is a major anabolic pathway that utilizes glucose to glucose-6 phosphate. Glucose metabolism via the pentose phosphate pathway is essential for producing NADPH. The deficiency of these enzymes has been linked to depression and psychotic disorders [45]. The pentose phosphate pathway is also a fundamental component of RNA and DNA precursors formation; it also produces reducing equivalents essential for glutathione pathway involved in DNA repair.

Metabolic pathways analysis proved the importance of nicotinate and nicotinamide metabolism in PFC of rats exposed to US. Nicotinamide (NAM) is a form of vitamin B3 (niacin). NAM in cells of human and animals is converted into nicotinamide mononucleotides and then into nicotinamide adenine dinucleotide (NAD+). Therefore, NAM is a key component of the metabolic pathway involved in NAD+ production [46,47]. NAM is essential for CNS neural cells’ development, life, and functioning, as it plays a role both in neuroprotection and neuronal death [48]. NAD+ plays an important role in aging and many pathological conditions [49]. NAM in high doses holds promise for treating a wide range of diseases, including neurological disorders, depression, and other mental problems. NAM is thought to have some antidepressant action by increasing levels of monoamine neurotransmitters, such as serotonin [46,50]. NAM’s bioaccessibility is crucial: a decrease in NAM concentration results in neurological deficit and dementia, while an excessive level of NAM is neurotoxic [48]. According to the results we obtained, there was a variation in NAM concentration levels in the frontal cortex of different groups. All experimental groups had lower NAM concentration levels compared to the control group in our study. However, there was no difference between the US group and groups who had treatment, which proved that stress induced by US exposure contributed to NAM deficiency in the frontal cortex, which could not be reversed by antidepressant treatment. It can be assumed that nicotinamide pathway disorders can play a role in pathogenesis of depression and also contribute to treatment failures.

Methionine concentration analysis in the PFC and hippocampus showed that methionine concentration decreased in all experimental groups compared against the control group. However, clomipramine administered intraperitoneally increased the concentration of methionine in the hippocampus compared to the US group and the nasal route of drug administration group. Nevertheless, there was no significant difference compared to healthy controls. Therefore, US stress exposure decreased methionine concentration in the cerebral cortex, which could not be reversed by antidepressant treatment and did not influence the hippocampus. Clomipramine administered intraperitoneally appeared to be effective in the hippocampus. Our data can contribute to a better understanding of the biochemical basis of depression. Methionine is an essential amino acid and is a precursor of succinyl-CoA, homocysteine, cysteine, creatine, and carnitine. Methionine regulates metabolic pathways in mammals. It is essential for polyamine, creatine, and phosphatidinecholine metabolism [51]. In the liver, l-methionine and adenosine-triphosphoric acid (ATP) synthesize S-Adenosyl methionine (SAM), using methionine adenosyltransferase. SAM is a coenzyme which plays a role in methyl transfer reactions, which in turn are widespread in all biological systems of the body and play a role in a number of metabolic processes [52]. According to the collected data, SAM has an antidepressant effect, so it can perform well as an antidepressant [53]. Nowadays, SAM is recommended as a second-line treatment if the first treatment given for a disease shows to be inefficient [54]. Therefore, in literature, there are proofs on methionine playing an important role in pathogenesis of depression, which is also proved by our data.

The aminoacyl tRNA biosynthesis pathway appeared to play a significant role in the PFC of animals exposed to US stress. Aminoacyl-tRNA synthetases play a key role in synthesis of proteins by linking amino acids to their cognate transfer RNAs. Aminoacyl-tRNA synthetases are enzymes universally distributed in the body, attaching the corresponding amino acid to its cognate tRNA [55,56]. ARSs participate in a variety of physiological and pathological processes through certain functions such as angiogenesis, post-translational modifications, translation initiation, and autophagy regulation [56]. Sources say that the Aminoacyl-tRNA biosynthesis pathway plays a significant role in the pathogenesis of depression. Yang et al. [57] demonstrated a two-sample Mendelian randomization analysis to assess the causal effects of 486 human serum metabolites on five major psychiatric disorders, including major depression. It was shown that Aminoacyl-tRNA biosynthesis is a significant metabolic pathway involved in major depression [57].

Metabolic pathways of arginine biosynthesis and arginine and proline metabolism occurred to be important in the PFC of rats exposed to US stress. Arginine, which is a semi-essential amino acid, is the substrate of major physiological processes in CNS, such as the urea cycle and nitric oxide cycle, so any imbalance of arginine can lead to metabolic disorders [58,59]. As has been shown in a large number of studies, there is a connection between imbalance of arginine blood concentration and accumulation of corresponding catabolic substances, NO regulation abnormalities, and the pathophysiology of major depressive disorder [60,61]. The study by Liu et al. [47] demonstrated that the arginine biosynthesis pathway plays a significant role in rats’ hippocampus after US-induced depressive-like behavior. Our study reaffirmed the significance of arginine metabolism in depression pathophysiology.

In contrast to the changes of the cortical metabolomic profile, we observed a difference of hippocampal metabolomic profiles between the control and animals exposed to the US stress, only in GABA and antioxidant protection. However, we observed differences in the group of rats with clomipramine injected intraperitoneally and the other groups in the purine metabolic pathway. Other authors also demonstrate changes in purine metabolism under the influence of antidepressants. The effect of the tricyclic antidepressant imipramine on purine metabolism has also been shown on mice in a model of depression [24]. It was demonstrated that purines play an important role in neuromodulation and neurotransmission, cellular-growth-related signaling, and energy metabolism [62]. There is a correlation found between purine metabolite concentration levels in spinal fluid and the concentration levels of monoamine metabolites such as homovanillic acid and 5-hydroxyindoleacetic acid, which proves a parallel metabolism of purines and monoamines in brain tissue [63]. It was shown that depressed patients had lower blood serum levels of inosine and guanosine and higher levels of xanthine in contrast to the control group [64]. In our study, there was a decrease in hippocampal hypoxanthine in the group of rats with clomipramine administered intraperitoneally.

Our study demonstrated the possible disturbances of antioxidant protection biochemical mechanisms which could probably result in oxidative damage in the brain tissue and finally neuronal cell death. Oxidative stress is caused by disturbances of pro-oxidant antioxidant balance, in favor of pro-oxidant [65]. Moderate oxidative stress plays a significant role in regulation of many biochemical processes in the body, but if the pro-oxidants are overproduced in the wrong place, this can lead to oxidative modification of proteins, lipids, and DNA [65,66]. Neuropsychiatric disorders, including depression, are linked to oxidative modification of nucleotides and multiple gene polymorphisms connected to the metabolism of reactive oxygen species. Oxidative stress plays a major role in pathophysiology of major depressive disorder, which is linked to free radical activity, stable molecules, and reactive oxygen and nitrogen species [65,67]. This was previously demonstrated in animal models of depression. In this way, depressive-like behavior in rats was linked to changes of oxidative stress marker levels in brain tissue, and in our study, we obtained the same results [68,69]. Rats become vulnerable to depression because of persistent oxidative stress, which could be reversed by antioxidant administration [70]. There are different protection systems and mechanisms in the body to cope with overproduction of pro-oxidants, but they can be intruded upon by different pathological states [65,67,71].

Our study, therefore, demonstrated that chronic exposure to variable frequency US induces depressive-like behavior and alters the metabolic profile of the frontal cortex and hippocampus in male rats in a certain way. It was observed that the cortical metabolome of the control group differed significantly from that of the other groups, which included changes in several pathways. These pathways are related to the glutamate, GABA, and tryptophan, which play a significant role in the pathogenesis of depression. The glutamate system plays a major role in the depression pathogenesis, and the decrease of glutamate and glutamine concentration levels, disruption of the glutamate–glutamine cycle, and GABA deficiency in the brain are biomarkers of a depressive-like state. We obtained the same findings in our research. We demonstrated the decrease of GABA in the two parts of a rat’s brain, and we also observed the decrease of glutamine concentration in the prefrontal cortex. These findings indicate abnormalities of glutaminergic neurotransmission, which can be a marker of an ultrasound-induced depressive state. The findings reinforce the construct validity of the ultrasound model of depression, and demonstrate that the model can be used to study depression. We also found changes in other metabolic pathways when exposed to variable US, such as the pentose phosphate pathway, nicotinate and nicotinamide metabolism, and glycine, serine, and threonine metabolism in the frontal cortex, which may provide additional information about possible mechanisms of depression. Regarding behavioral and metabolomic data in the body, it should be noted that groups of rats with different routes of clomipramine administration behaved similarly, so both routes of administration were of the same efficiency. Interestingly, in our study, clomipramine affected global metabolic profiles in the frontal cortex and hippocampus in a different way that depended on the route of clomipramine administration. The intranasal route of administration was associated with more significant changes of metabolite composition in the frontal cortex compared to the control and US groups, while the intraperitoneal route of administration corresponded with more profound changes in hippocampal metabolome compared to all other animal groups.

## 4. Materials and Methods

### 4.1. Animals

We used male Wistar rats from the Nursery for Laboratory Animals (Pushchino, RAS, Moscow region) in the experiment that were 2 months old (200–250 g). All animals were kept at a constant temperature (23 °C) with controlled direct lighting (12/12 h) and had free access to water and food. Animals were housed individually in polycarbonate transparent cages (42 × 26 × 15 cm) during stress protocols. After the stress protocol, rats were placed in cages in groups. Housing conditions and all experimental procedures were set up and maintained in accordance with Directive 2010/63/EU of 22 September 2010 and approved by the local ethical committee of V.P. Serbsky National Medical Research Center for Psychiatry and Narcology.

### 4.2. Experimental Groups and Procedures

The animals were divided into the following experimental groups: control animals (*n* = 8); rats exposed to ultrasonic radiation for 3 weeks (*n* = 8); and rats exposed to ultrasonic radiation for 3 weeks with clomipramine intranasal (*n* = 8) and intraperitoneal treatment (*n* = 8). On the next day after US exposure, the sucrose preference test was performed. The tests were conducted with a delay of 1 day between them in the following order: social interaction test, forced swim, and Morris water maze.

The US exposure was performed for 24 h each day for 3 weeks and consisted of periods within the following ranges: low frequencies (20–25 kHz), middle range frequencies (25 < x < 40 kHz), and frequencies of high range (40–45 kHz). The US frequencies changed every 10 min. Low and middle frequency US constituted 35% of emissions each; high frequencies constituted 30% of emission time. The loudness of the sound fluctuated at the range ±10% of the averaged value, i.e., 50 ± 5 dB. The ultrasonic device (Weitech, Belgium) was suspended from the ceiling, and the loudspeaker was oriented downwards, where there were cages with rats at a distance of 2 m. The position of the cages was changed every 3 days.

Clomipramine (Sigma-Aldrich, St. Louis, MO, USA) was administered daily at 10 am intranasally or intraperitoneally at a dose of 7.5 mg/kg for the duration of the ultrasound exposure from days 1 to 21. IP Clomipramine was administered in a volume of 500 μL, IN in a volume of 30 μL. Each rat was gently held with the ventral side up and the drug was pipetted with 5 μL delivered to each nostril (total three times in each nostril). The US and control groups received intranasal saline in a volume of 30 μL for 21 days.

### 4.3. Behaviour Test

#### 4.3.1. Social Interaction Test

The social interaction test was performed as described previously [13]. A juvenile male, a 4 weeks of age non-bred white rat, was placed into the home cage of the tested Wistar rat for 10 min. Their behavior was video recorded. The duration of social interaction was defined as any proactive contacts that the Wistar rat showed toward the juvenile male, which was typically composed of approaches, body contacts, following, sniffing, and exploration, and was scored using RealTimer software (OpenScience, Moscow, Russia).

#### 4.3.2. Sucrose Preference Test

During this test, rats or mice were given, for 24 h, a free choice between two bottles, one with a 1% sucrose solution and another with tap water. At the beginning and end of the test, the bottles were weighed and consumption was calculated. The beginning of the test started with the onset of the dark (active) phase of the animals’ cycle. To prevent the possible effects of side-preference in drinking behavior, the position of the bottles in the cage was switched every 6 h. No previous food or water deprivation was applied before the test. The percentage of preference for sucrose was calculated using the following formula: Sucrose Preference = Volume (Sucrose Solution)/(Volume (Sucrose solution) + Volume (Water)) × 100.

#### 4.3.3. Forced Swim Test

A transparent round pool made of plastic (diameter—31 cm, height—40 cm) was filled with water (temperature of 23 °C) to a level that prevented a rat from touching the bottom and was lit with weak light (15 lx). Animals were introduced for 8 min to the pool. The duration of floating, determined as the absence of any directed movements of the animals’ head and body, was video-recorded—and scored off-line during the last 6 min of the test—using a digital camera, and was scored using RealTimer software (OpenScience, Moscow, Russia) [13].

#### 4.3.4. Morris Water Maze Test

The Morris labyrinth was a grey-walled pool 150 cm in diameter and 60 cm high, filled with 40 cm of water. The water temperature was 230 °C. An 8-cm-diameter round platform made of plastic to match the color of the unit was placed in the center of one of the quadrants of the pool, 2 cm below the surface of the water. The pool was placed in a room with many spatial landmarks. The test was conducted under room light. Animals were trained for one session, during which they were placed in the water from 8 different points at approximately the same distance from the platform. After an animal reached the platform, it was left on the platform for 15 s, then placed in a separate cage for 60 s. Rats that did not find the platform within 60 s were gently guided towards it. In each trial, the time required to reach the platform was recorded. After 48 h, long-term memory was assessed in a single trial. The location of the environmental stimuli and the platform was kept constant throughout the experiment. This test was also chosen last because it is the most stressful [14].

### 4.4. Tissue Sample Collection and Pretreatment

Once all the behavioral tests were completed, the rats were anaesthetized with isoflurane (4%, 2 L/min) in a chamber until their toe reflex disappeared. The rats were then quickly sacrificed by decapitation. The FC and hippocampus were removed from the brain, then quickly frozen with liquid nitrogen, and finally stored at −80 °C before use. The hippocampus and prefrontal cortex were isolated as described elsewhere [70].

### 4.5. Sample Analysis

Before the analysis, a previously weighed sample was filled into a labeled vial. The BeadBeater balls (200 µL) and 250 µL of internal standard solution were then added. The contents were homogenized (BeadBeater) for 5 min. The sample was then centrifuged at 1000 g for 1 min. An amount of 150 µL of liquid was taken by the Eppendorf pipette and added to 100 µL of chloroform. The sample was placed on Vortex for 5 min, then 50 µL of methanol cooled to −80 °C was added. The sample was placed on Vortex for 2 min and then centrifuged at maximum speed for 10 min at 4 °C. After the centrifugation, 100 µL of liquid layer was taken and filled into labeled vials with inserts and brought to analysis.

Metabolic analysis was performed on a SCIEX 4500 QTRAP combined with Shimadzu CBM-30ACMP liquid chromatograph. Each compound was identified using LC/MS/MS in the mode of multiple-reaction monitoring (MRM). The compound was regarded as faithfully identified if the results of the analysis met the criteria of sensitivity and specificity: signal-to-noise ratio less than 5; at least two MRM transitions for each substance were monitored (glycine and proline excluded); MRM transition intensities variation being no more than 20% of possible values; mean holding time deviation of chromatographic peaks no more than 5%. Evaluation of the chromatographic system stability was made using analytic standards for analyzed substances (Table 3).

Chromatographic separation parameters were the following: Buffer A (deionized water, 10 mM ammonium acetate); buffer B (90% acetonitrile, 10% water, 10 mM ammonium acetate); the chromatography programme was as follows: 0 min 99% B, 0.5 min 99% B, 3 min 75% B, 9 min 50% B, 11.5 min 50% A, 12 min 99% B; 14 min 99% B; flow 0.8 mL/min; thermostat temperature was 40 °C. For chromatographic separation of the substances, we used a chromatography column SeQuant ZIC-HILIC PEEK 3.5 µm, 100 Å 150 × 4.6 mm.

### 4.6. Statistical Analysis

Analysis was performed using XLSTAT software [72]. (accessed on 3 September 2021). All groups in all tests were tested for normal distribution by Shapiro–Wilk test. One-way ANOVA followed by Tukey’s post hoc test were then applied to compare these groups. Kruskal–Wallis test was used to compare groups in the Morris test. The alpha level was set as *p* < 0.05. ANOVA was used to compare groups when comparing the metabolites with FDR correction for multiple comparison. Descriptive data was presented as Mean ± SD if other is not specified.

Correlation-Based Principal Component Analysis (PCA) was performed on metabolites in the hippocampus and frontal cortex (FC). Biplots of the first 2 principal components were built with both metabolite factor loadings and observation principal component scores by group. Squared average distance was used as the scaling factor for the biplot. After that, in order to check differences between animal groups, the principal components 1 and 2 scores for every animal group were entered in the linear regression model to compare slopes of regression lines.

To visualize between-group differences, clustered heatmaps were created using Ward’s Agglomerative Hierarchical Clustering of metabolites of animal groups for both individual animal values and the group. Means were calculated based on Euclidean dissimilarity distance after normalization and centering values on a scale from −1 to +1. The heat maps of the mean values were also clustered by groups using the K-means method.

The metabolic pathways were constructed and analyzed using the Pathway Analysis based on MetaboAnalyst 5.0 platform and Kyoto Encyclopedia of Genes and Genomes (KEGG), to identify the potential affected biochemical pathways and visualize metabolic networks. The pathway library chosen was *Rattus norvegicus* (rat). Pathways with *p*-value < 0.05 were considered to be significantly altered. The pathways with the Impact value > 0.1 were considered as the most relevant pathways involved in depression-like behavior and clozapine effects.

## 5. Conclusions

The stress induced by exposure to variable frequency ultrasound (US) in a three-week period caused depressive-like behavior in rats. Clomipramine administration has antidepressant effects, and there was no difference observed in the efficacy of intranasal and intraperitoneal administration.

US stress contributed to glutamate system disorders, imbalance in the antioxidant defense system, nicotinamide deficiency, and metabolic disorders of proline and tryptophan. GABA was disrupted in the frontal cortex and hippocampus in the ultrasound groups.

As for the metabolomic profile, the differences between the two routes of administration were identified in alanine, aspartate, and glutamate pathways in the frontal cortex and purine metabolism in the hippocampus.

In summary, our study demonstrated that exposure to variable frequency US contributed to changes of the metabolomic pathways associated with pathophysiology of depression.

The efficacy of clomipramine was demonstrated in behavioral tests. Despite the fact that there were no differences between the two modes of clomipramine administration, we observed metabolomic profile changes.

Before the present study, no analysis of metabolomic profile had been performed for different routes of drug administration. Since far metabolic processes in brain tissue can change in many ways depending on different routes of antidepressant administration, the antidepressant therapy should also be evaluated from this point of view.

## Figures and Tables

**Figure 1 ijms-22-09598-f001:**
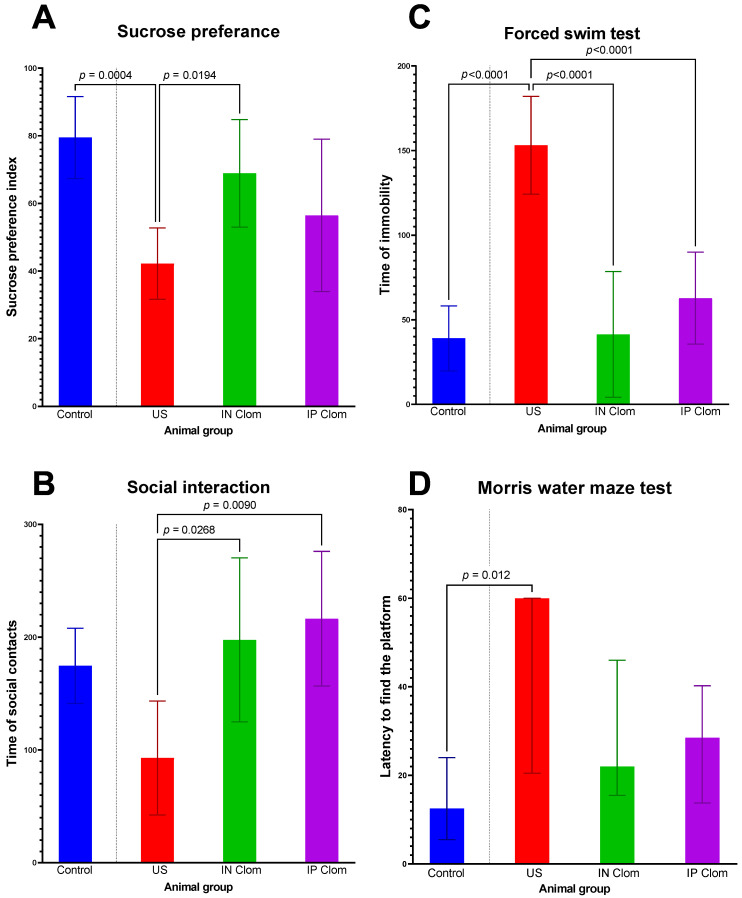
Results of the behavioral tests in the US rat model of depression. (**A**)—Sucrose preference test, (**B**)—social interaction test, (**C**)—forced swim test, (**D**)—Morris water maze test. IN Clom—clomipramine intranasal administration, IP Clom—clomipramine intraperitoneal administration; US—ultrasound stress without treatment. The Morris water maze test median and interquartile range are presented and a Kruskal–Wallis test with post-hoc Dunn’s multiple comparisons test was used. For other groups, means and 95% confidence intervals are presented and ANOVA with post-hoc Tukey’s multiple comparisons test was used.

**Figure 2 ijms-22-09598-f002:**
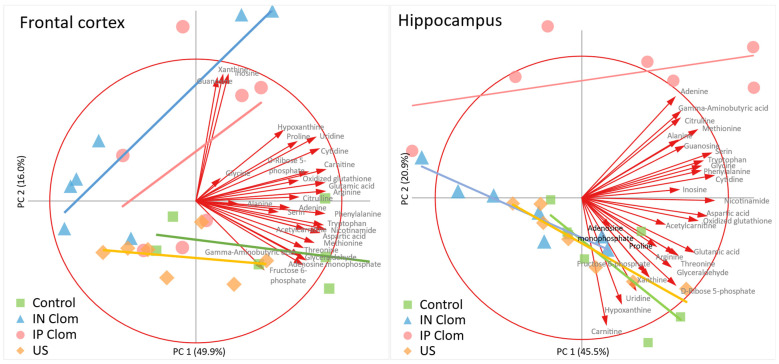
Principal component analysis biplots of metabolite component scores and animals factor loadings by group on principal component 1 (PC1) and principal component 2 (PC2) for the frontal cortex (**left**) and hippocampus (**right**). Principal components 1 and 2 explain 66% of metabolite content variability in both the frontal cortex and hippocampus. Bold lines are the linear regression lines for individual principal component coordinates for each animal group. Arrows represent vectors of metabolite factor scores. Points represent principal component scores for individual animals. IN Clom—clomipramine intranasal administration, IP Clom—clomipramine intraperitoneal administration; US—ultrasound stress without treatment. The PCA result indicated that metabolic profiles differ in the control and US groups from both clomipramine groups in the frontal cortex and in the intraperitoneal clomipramine group from every other group in the hippocampus. These differences were confirmed with MANOVA analysis of PCA coordinates (Appendix A).

**Figure 3 ijms-22-09598-f003:**
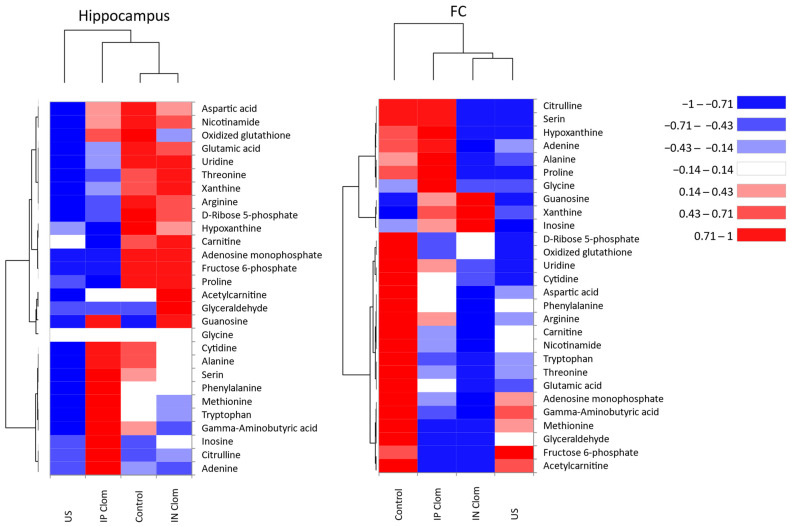
Heatmaps of metabolomics data for the hippocampus (left) and frontal cortex (right). The color of each section is proportional to the metabolite mean concentration in the animal group normalized and centered on the scale from −1 to +1. Rows represent metabolites and columns represent animal groups. The dendrograms based on the Euclidean distances accompany the heatmaps; IN Clom—clomipramine intranasal administration, IP Clom—clomipramine intraperitoneal administration; US—US stress without treatment. Grouped heatmaps of metabolic data in individual animals can be seen in Appendix A.

**Figure 4 ijms-22-09598-f004:**
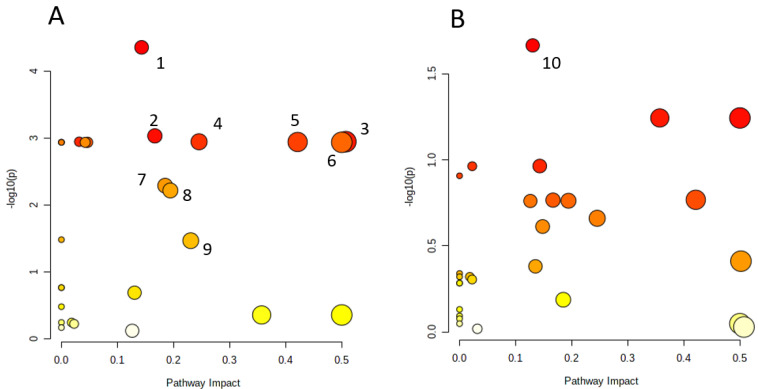
Pathway analysis of metabolites. (**A**). Pathway analysis of metabolites in frontal cortex related to depression-like behaviors induced by ultrasound stress. (**B**). Pathway analysis of metabolites in the hippocampus affected by the antidepressant effect of intraperitoneal clomipramine; 1—Tryptophan metabolism; 2—Aminoacyl-tRNA biosynthesis; 3—Alanine, aspartate, and glutamate metabolism; 4—Arginine and proline metabolism; 5—Arginine biosynthesis; 6—d-Glutamine and d-glutamate metabolism; 7—Pentose phosphate pathway; 8—Nicotinate and nicotinamide metabolism; 9—Glycine, serine, and threonine metabolism; 10—Purine metabolism. The size and color of each circle was based on pathway impact value and *P*-value, respectively.

**Table 1 ijms-22-09598-t001:** Concentration of metabolites in the frontal cortex (FC) and hippocampus (Hipp) of rats after ultrasound stress. IN Clom—Clomipramine intranasal administration, IP Clom—Clomipramine intraperitoneal administration; US—US stress without treatment. All groups were compared using the Fisher test. FDR was performed for multiple comparison correction. The results of the post hoc analysis are in supplementary materials (Appendix A). The bold font shows statistical differences.

Metabolites	PubChem CIDs	Structure	Control	IN Clom	IP Clom	US	F and *p*-Value	FDR
Proline	614	Hipp	0.97 ± 0.13	1.08 ± 0.28	0.83 ± 0.26	1.002 ± 0.2	F = 1.5 *p* = 0.24	0.310
		FC	0.852 ± 0.09	0.7 ± 0.21	0.95 ± 0.21	0.69 ± 0.1	F = 3.9 *p* = 0.02	**0.039**
Nicotinamide	936	Hipp	4.05 ± 0.32	3.54 ± 0.57	4.01 ± 0.99	3.96 ± 0.45	F = 0.96 *p* = 0.43	0.470
		FC	4.13 ± 0.54	2.81 ± 0.53	3.26 ± 0.34	3.42 ± 0.42	F = 9.5 *p* = 0.0003	**0.002**
Phenylalanine	994	Hipp	0.32 ± 0.06	0.25 ± 0.071	0.41 ± 0.24	0.33 ± 0.09	F = 1.58 *p* = 0.22	0.310
		FC	0.31 ± 0.12	0.17 ± 0.051	0.23 ± 0.04	0.23 ± 0.068	F = 3.6 *p* = 0.03	0.050
Alanine	602	Hipp	0.14 ± 0.011	0.12 ± 0.027	0.17 ± 0.06	0.14 ± 0.017	F = 2.4 *p* = 0.09	0.182
		FC	0.11 ± 0.037	0.08 ± 0.018	0.14 ± 0.016	0.09 ± 0.02	F = 7.3 *p* = 0.001	**0.005**
Adenine	190	Hipp	0.1 ± 0.01	0.09 ± 0.021	0.18 ± 0.054	0.1 ± 0.011	F = 13.14 *p* < 0.001	**<0.001**
		FC	0.09 ± 0.012	0.07 ± 0.013	0.091 ± 0.012	0.08 ± 0.014	F = 3.17 *p* = 0.043	0.066
Serine	617	Hipp	0.35 ± 0.057	0.28 ± 0.067	0.44 ± 0.15	0.35 ± 0.064	F = 3.4 *p* = 0.033	0.110
		FC	0.31 ± 0.06	0.23 ± 0.022	0.32 ± 0.05	0.24 ± 0.025	F = 8.5 *p* = 0.0005	**0.003**
Glyceraldehyde	751	Hipp	0.027 ± 0.019	0.015 ± 0.005	0.016 ± 0.015	0.028 ± 0.015	F = 1.5 *p* = 0.24	0.310
		FC	0.015 ± 0.009	0.003 ± 0.002	0.003 ± 0.002	0.008 ± 0.005	F = 7.34 *p* = 0.001	**0.005**
Cytidine	6175	Hipp	1.78 ± 0.34	1.34 ± 0.35	1.91 ± 0.64	1.58 ± 0.26	F = 2.42 *p* = 0.091	0.182
		FC	1.16 ± 0.29	0.92 ± 0.41	1.01 ± 0.18	0.89 ± 0.12	F = 1.4 *p* = 0.27	0.282
Acetylcarnitine	7045767	Hipp	0.94 ± 0.21	0.83 ± 0.19	0.99 ± 0.32	1.1 ± 0.25	F = 1.5 *p* = 0.24	0.310
		FC	0.53 ± 0.17	0.39 ± 0.15	0.39 ± 0.12	0.5 ± 0.07	F = 2 *p* = 0.14	0.170
Carnitine	288	Hipp	2.21 ± 0.29	1.8 ± 0.37	0.66 ± 0.16	2.13 ± 0.52	F = 27.53 *p* < 0.001	**<0.001**
		FC	1.12 ± 0.18	0.83 ± 0.33	0.92 ± 0.27	0.94 ± 0.19	F = 1.6 *p* = 0.22	0.250
Xanthine	1188	Hipp	0.15 ± 0.06	0.09 ± 0.061	0.09 ± 0.023	0.14 ± 0.047	F = 2.5 *p* = 0.08	0.182
		FC	0.02 ± 0.009	0.043 ± 0.03	0.038 ± 0.027	0.024 ± 0.006	F = 2 *p* = 0.14	0.170
l-glutation oxydazed	65359	Hipp	0.2 ± 0.05	0.11 ± 0.052	0.16 ± 0.098	0.14 ± 0.018	F = 2.7 *p* = 0.07	0.182
		FC	0.11 ± 0.03	0.081 ± 0.042	0.07 ± 0.023	0.06 ± 0.02	F = 3.5 *p* = 0.032	0.053
Uridine	6029	Hipp	0.05 ± 0.01	0.04 ± 0.015	0.037 ± 0.012	0.05 ± 0.012	F = 1.87 *p* = 0.16	0.270
		FC	0.028 ± 0.005	0.022 ± 0.009	0.025 ± 0.008	0.02 ± 0.005	F = 1.4 *p* = 0.28	0.282
Ribose 5-phosphate	77982	Hipp	0.07 ± 0.031	0.026 ± 0.013	0.029 ± 0.02	0.05 ± 0.012	F = 6.5 *p* = 0.002	**0.010**
		FC	0.027 ± 0.007	0.017 ± 0.011	0.014 ± 0.005	0.013 ± 0.009	F = 4 *p* = 0.018	**0.039**
Threonine	205	Hipp	1.24 ± 0.21	1.04 ± 0.28	0.96 ± 0.32	1.14 ± 0.27	F = 1.4 0.28	0.336
		FC	1.39 ± 0.37	0.9 ± 0.132	0.98 ± 0.14	0.97 ± 0.21	F = 6.4 0.002	**0.008**
Glycine	750	Hipp	0.0018 ± 0.0003	0.0015 ± 0.0005	0.0021 ± 0.001	0.0017 ± 0.0004	F = 0.8 *p* = 0.49	0.510
		FC	0.0012 ± 0.0004	0.001 ± 0.0003	0.0017 ± 0.0003	0.0011 ± 0.0002	F = 4.4 *p* = 0.013	**0.038**
Fructose 6-phosphate	69507	Hipp	0.0085 ± 0.009	0.002 ± 0.0016	0.003 ± 0.003	0.0068 ± 0.004	F = 2.17 *p* = 0.12	0.220
		FC	0.0021 ± 0.0009	0.00078 ± 0.0002	0.00078 ± 0.0009	0.0024 ± 0.002	F = 4.25 *p* = 0.015	**0.039**
Citrulline	833	Hipp	0.58 ± 0.14	0.65 ± 0.17	1.1 ± 0.4	0.645 ± 0.14	F = 7 *p* = 0.002	**0.008**
		FC	0.44 ± 0.14	0.33 ± 0.08	0.45 ± 0.087	0.335 ± 0.09	F = 2.81 *p* = 0.06	0.088
GABA	119	Hipp	2.35 ± 0.52	1.15 ± 0.29	3.62 ± 0.98	1.5 ± 0.32	F = 23.7 *p* < 0.001	**<0.001**
		FC	1.71 ± 0.4	0.74 ± 0.18	0.97 ± 0.22	1.41 ± 0.2	F = 18.55 *p* < 0.001	**<0.001**
Tryptophan	1148	Hipp	0.5 ± 0.11	0.42 ± 0.13	0.7 ± 0.42	0.47 ± 0.17	F = 1.75 *p* = 0.18	0.280
		FC	0.58 ± 0.16	0.31 ± 0.11	0.33 ± 0.05	0.37 ± 0.098	F = 8.6 *p* = 0.0005	**0.003**
Guanosine	135398635	Hipp	0.005 ± 0.002	0.0037 ± 0.002	0.0082 ± 0.0045	0.005 ± 0.001	F = 3.3 *p* = 0.04	0.120
		FC	0.0013 ± 0.0005	0.004 ± 0.0048	0.003 ± 0.002	0.0011 ± 0.0005	F = 2.2 *p* = 0.11	0.147
AMP	6083	Hipp	1.75 ± 0.57	1.98 ± 0.88	1.5 ± 0.8	1.88 ± 0.59	F = 0.57 *p* = 0.64	0.641
		FC	2.84 ± 0.46	1.61 ± 0.61	2.06 ± 0.77	2.4 ± 0.43	F = 5.5 *p* = 0.005	**0.015**
Arginine	232	Hipp	0.21 ± 0.03	0.15 ± 0.05	0.15 ± 0.07	0.196 ± 0.045	F = 2.5 *p* = 0.08	0.182
		FC	0.2 ± 0.05	0.15 ± 0.057	0.18 ± 0.033	0.174 ± 0.038	F = 1.35 *p* = 0.28	0.282
Aspartic acid	424	Hipp	0.36 ± 0.05	0.25 ± 0.065	0.32 ± 0.16	0.35 ± 0.065	F = 1.9 *p* = 0.15	0.260
		FC	0.39 ± 0.09	0.26 ± 0.072	0.33 ± 0.05	0.31 ± 0.074	F = 3.98 *p* = 0.02	**0.039**
Glutaminic acid	611	Hipp	4.89 ± 0.4	4.24 ± 1.01	4.24 ± 1.05	4.66 ± 0.93	F = 0.94 *p* = 0.44	0.470
		FC	4.76 ± 0.71	3.4 ± 1.23	3.83 ± 0.46	3.63 ± 0.45	F = 4 *p* = 0.02	**0.039**
Methionine	876	Hipp	0.1 ± 0.017	0.06 ± 0.018	0.17 ± 0.1	0.088 ± 0.014	F = 5.88 *p* = 0.004	**0.015**
		FC	0.087 ± 0.02	0.038 ± 0.014	0.039 ± 0.011	0.062 ± 0.015	F = 14 *p* < 0.001	**<0.001**
Ionosine	135398641	Hipp	0.059 ± 0.05	0.04 ± 0.018	0.068 ± 0.027	0.046 ± 0.019	F = 1.2 *p* = 0.32	0.380
		FC	0.015 ± 0.007	0.025 ± 0.016	0.019 ± 0.014	0.008 ± 0.004	F = 2.5 *p* = 0.08	0.112
Hypoxanthine	135398638	Hipp	0.17 ± 0.06	0.14 ± 0.035	0.067 ± 0.022	0.14 ± 0.018	F = 9.4 *p* = 0.0003	**0.002**
		FC	0.127 ± 0.02	0.11 ± 0.022	0.13 ± 0.045	0.103 ± 0.03	F = 1.55 *p* = 0.23	0.254

**Table 2 ijms-22-09598-t002:** Results of a pairwise comparison of regression slopes for the principal scores on the first two PCA dimensions between the animal groups for the frontal cortex and hippocampus.

Brain Structure		Frontal Cortex				Hippocampus			
Group		Control	IN Clom	IP Clom	US	Control	IN Clom	IP Clom	US
Control	SE	-	0.15	0.339	0.128	-	0.171	0.207	0.181
	t	-	4.262	1.501	0.181	-	−1.471	−3.218	−0.939
	df	-	12	12	12	-	12	12	12
	*p*	-	0.001 *	0.159	0.86	-	0.167	0.007 *	0.366
IN Clom	SE	0.15	-	0.361	0.169	0.171	-	0.121	0.072
	t	4.262	-	−0.364	−3.645	−1.471	-	−3.432	1.129
	df	12	-	12	12	12	-	12	12
	*p*	0.001 *	-	0.722	0.003 *	0.167	-	0.005 *	0.281
IP Clom	SE	0.339	0.361	-	0.37	0.207	0.121	-	0.137
	t	1.501	−0.364	-	−1.315	−3.218	−3.432	-	3.628
	df	12	12	-	12	12	12	-	12
	*p*	0.159	0.722	-	0.213	0.007 *	0.005 *	-	0.003 *
US	SE	0.128	0.169	0.37	-	0.181	0.072	0.137	-
	t	0.181	−3.645	−1.315	-	−0.939	1.129	3.628	-
	df	12	12	12	-	12	12	12	-
	*p*	0.86	0.003 *	0.213	-	0.366	0.281	0.003 *	-

Note: *—statistically significant differences between regression slopes at *p* < 0.05.

**Table 3 ijms-22-09598-t003:** The list of multiple-reaction monitoring transitions for the analyzed substances.

Substance	Q1	Q3		Polarity
l-Glutathione oxidized	613.008	354.9	484	+
Adenine	135.93	119	92	+
Adenosinmonophosphat	347.971	136	119	+
Alanine	89.988	44.1	N/A	+
Arginine	175.074	70	116	+
Aspartic acid	133.921	74	88	+
γ-Aminobutyric acid	104.076	87	69	+
Hypoxanthine	136.97	119	94	+
Glyceraldehyde 3-phosphate	168.748	97	78.9	−
Glycine	75.985	30	48	+
Guanosine	281.834	150	133	−
Inosine	266.762	134.9	134.5	−
Carnitine	162.011	103	60.1	+
Xanthine	150.833	107.9	42	−
Methionine	150.006	104	133	+
Nicotinamide	122.99	80	78	+
*O*-Acetyl-l-carnitine	204.085	85	145.1	+
Proline	116.03	70.1	N/A	+
Ribose 5-phosphate	228.69	96.9	78.9	−
Serine	105.982	60	42	+
Threonine	119.969	74	102	+
Tryptophan	205.01	188.1	145.9	+
Uridine	242.799	199.8	110	−
Phenylalanine	166.011	119.9	102.9	+
Citidine	244.016	112	95	+
Citrulline	176.021	158.9	113.1	+
Cotinine-d3	180.04	85	101	+

## Data Availability

Data available on request.

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
