# Peer review of "Brain Metabolic Profile after Intranasal vs. Intraperitoneal Clomipramine Treatment in Rats with Ultrasound Model of Depression"

_ijms, 2021, doi:10.3390/ijms22179598_

Round 1

Reviewer 1 Report

The manuscript entitled “Brain metabolic profile after intranasal vs. intraperitoneal clomipramine treatment in rats with ultrasound model of depression” by Abromava and colleagues aims to investigate the behavioural phenotypes and brain metabolic profile of a newly established animal model of depression. This animal model is based on the exposure of rats to unpredictably alternating frequencies of ultrasound for 3 weeks. This model has been shown to produce depression-like phenotypes, including anhedonia-like behaviours, passive stress-coping behaviour, deficits in locomotion and social behaviours. Such phenotypes were reversed by administration of the antidepressant fluoxetine and accompanied with major changes, at the molecular level, in the prefrontal cortex and hippocampus, two brain areas associated with depression.

In the present manuscript, the authors confirm the behavioural phenotypes produced by exposure to the ultrasound and determine i) whether they could be reversed by the antidepressant clomipramine, ii) whether the effects of clomipramine depend on the route of administration (intranasal versus intraperitoneal) and iii) whether they were accompanied by changes in metabolic profiles in the prefrontal cortex and hippocampus.

Although the study is original, rather well conducted and very relevant to depression, the manuscript itself is not well written and poorly structured. The English scientific writing is also of low quality for some parts of the text.

I suggest the authors provide major revisions to their manuscript before submitting it again to the journal. Below are a few points to consider before re-submission:

  • The abstract is poorly written (structure and language) and does not contain any background information/relevance of the study. In addition, there are too many details given regarding the pathways and there is no interpretation of the findings as well as no conclusion/perspectives.
  • The introduction is rather well written and the aims are clearly stated. A better justification should be provided, however, regarding the use of ultrasound for producing phenotypes relevant to depression. More details regarding the published data on the model should also be provided (behaviours, brain aspects…).
  • The Method section is well written for the molecular part but not for the behavioural part. 
    • More details (and references to other papers) should be provided regarding the behavioural assays. The authors do not provide enough information to understand how the behavioural tests were precisely conducted (sex and age of the animal? lighting conditions? habituation to two bottles and measure of water intake in the sucrose preference test? why no pretest in the forced swim test? why only 8 min for the forced swim test (duration not conventional), references for each behavioural test...). 
    • Details should be provided regarding the antidepressant administration: dose, volume of injection, timing of injection before the behavioural tests, origin of purchase…
    • Details should be provided regarding the possible anesthetic method used during the sacrifice (decapitation) of the animals.
    • The authors should be consistent in using ”frontal cortex“ or “prefrontal cortex” in the whole manuscript, as the prefrontal cortex is only a part of the frontal cortex. By the way, how did the authors harvest precisely the frontal cortex/prefrontal cortex and hippocampus of the rats? It is important to precise the anatomical boundaries of these two brain regions.
    • Avoid abbreviations in the titles of the figures/tables and define abbreviations used in the legend.
    • Table 3 (pathway analysis) may be put into the supplement. It contains redundant data with Figure 4.
    • Why did the authors not characterize the hypothalamic–pituitary–adrenal axis activity and reactivity? Alterations in the hypothalamic–pituitary–adrenal axis is considered an endophenotype of depression and found in many animal models of depression. 
    • Why did the authors not inject the vehicle to the US rats, to be comparable with the US rats receiving Clom IP or IN??
  • The numbering of the figures and tables should be checked (there is 2 Tbales 3) and the referencing in the main text to the tables and figures should be adjusted accordingly. 
  • Please check how to report the outcomes of the ANOVA test and the pairwise comparison in the whole Result section!
  • The results of the behavioural tests are very poorly written. 
    • It would be better to describe test by test and write a short conclusion of the findings, yet without interpretation.
    • Why represent behavioural data as box plots (which should be defined) when conducting parametric analyses? 
    • Please choose different symbols for the pairwise comparisons : ultrasound versus control rats, clomipramine versus ultrasound rats and ultrasound versus control rats. Please check that the symbols are positioned correctly above the graph bars.
      • The symbol # above the IP Clom in the sucrose preference test does not look at the right place as it is unlikely that the US and the IP Clom significantly differed in the sucrose index (visually speaking).
    • Homogenize the use of Porsolt or Forced Swim test.
    • It would be good to separate Table 1 into 2 tables, one for the prefrontal cortex and one for the hippocampus as Table 1 is very big. Also, it would be good to use the same symbols as in the Figure 1 to highlight the pairwise comparisons (the p value does not indicate to the reader which groups are significantly different).
  • Not enough details are provided for Figure 2.
    • In Figure 2 and 3, why using Hippocamp as an abbreviation for hippocampus??
  • The Table 2 contains statistical outputs and should be in the supplement.
  • Line 182: it cannot be postdoc ANOVA, please correct.
  • The heat-maps Figure 3 is the representation of Table 1, somehow I do not catch the differences and how the normalization of the data from Table 1 to Figure 3 was performed.
  • A better explanation of the choice of the metabolic pathways by brain region from the results in Table 1 should be provided.
  • The justification of the method used lines 211-213 should be put in the Method section.
  • The Discussion section is very lengthy and definitely lacks a good structure. It should be rewritten to highlight the aims and results of the study, and then provide some interpretation of the data as well as limitations and perspectives. The Discussion provided by the authors is only reflecting, to my opinion, text-book knowledge about the different metabolic pathways altered in the rat model of depression.
    • The Discussion section should be entirely rewritten to summarize the results and interpret them regarding the existing literature and possible implications for our understanding of major depression.
  • The use of the forced swim test should be referred to as stress-coping and not “depression”-like behaviours, please see papers PMID: 28287253 and PMID: 30738104.
  • The alterations in mitochondria in depression have been recently reviewed in PMID: 32979495.
  • Please make sure to define all abbreviations before first mention and avoid starting a sentence with an abbreviation or a number.
  • Make sure to homogeneize the past tense in the Method and Result sections.

Author Response

We thank the Reviewer 1 for such complete and detailed analysis of our work and helpful recommendations. All the recommendations of this reviewer were implemented in the manuscript. Together, we feel that the suggestions and questions of Reviewer 2 have substantially contributed in the improvement of our work and would like to express our gratitude for this important input.

We presented our paper corrections in more understandable way according to yours recommendations. For cross-correction procedure simplicity we numbered lines and pages in the text and specified corresponding numbers for commentary to questions. We have highlighted in yellow the corrections in the text. We hope that such form of answer will be appropriate.

Answer on questions from Reviewer 1: 

  • The abstract is poorly written (structure and language) and does not contain any background information/relevance of the study. In addition, there are too many details given regarding the pathways and there is no interpretation of the findings as well as no conclusion/perspectives.

We fully agree with this highly important point rose by Reviewer 1 and thankful to him/her for bringing this question up. We have corrected the abstract in accordance with the requests of Reviewer 1.

  • The introduction is rather well written and the aims are clearly stated. A better justification should be provided, however, regarding the use of ultrasound for producing phenotypes relevant to depression. More details regarding the published data on the model should also be provided (behaviours, brain aspects…).

Following a recommendation of Reviewer 1, we added a few sentences in introduction (Page 2, line 91-95). Here we would like to point out that alterations in the hypothalamic-pituitary-adrenal axis are observed in depression and shown also in animal models. In the US model of depression, the level of epinephrine, norepinephrine and corticosterone was elevated after US stress. The decreased levels of serotonin, dopamine and its metabolites were observed in blood.

  • The Method section is well written for the molecular part but not for the behavioural part. 
    • More details (and references to other papers) should be provided regarding the behavioural assays. The authors do not provide enough information to understand how the behavioural tests were precisely conducted (sex and age of the animal? lighting conditions? habituation to two bottles and measure of water intake in the sucrose preference test? why no pretest in the forced swim test? why only 8 min for the forced swim test (duration not conventional), references for each behavioural test...). 

We are thankful to Reviewer 1 for this observations.

We added information to Materials and Methods on the sex and age of animals. Information about the lighting conditions is in the manuscript (Page 16, line 515-520).

We have also described each behavioral test in more detail (Page 17, line 546-584).

As for the 8 minutes in the forced swim test, given that during the experiment the animals were tested using multiple models, we tried to choose the least stressful test options so that the testing procedure itself would have minimal on the condition of the rats. On the basis of the literature data, the duration of the of the forced swimming test, only slightly exceeding the minimum time for rats (Bourin M, Mocaër E, Porsolt R. Antidepressant-like activity of S20098 (agomelatine) in the forced swimming test in rodents: involvement of melatonin and serotonin receptors. J Psychiatry Neurosci. 2004 Mar;29(2):126-33,article attached, p. 128)

The pretest in the forced swim test: We used, as some authors describe – animal friendly Porsolt version (Gregus 2005), and we decided, according to this article, that if we have prior stress, to do the test in a single trial. In the traditional forced swim test developed by Porsolt, rats are placed into a swim tank for 15 min on day 1 to induce a state of “helplessness”. The rats are then placed back into the swim tank on day 2 for 10 min, during which behaviors indicative of depression are measured. In this experiment, each rat was tested in the forced swim tank once only. This was done because we sought to assess “helplessness” as induced by prior exposure to either repeated restraint stress or repeated corticosterone injections. Armario (2021) criticized the pretest in rats because perhaps immobility in the test itself reflects more adaptation to conditions rather than a stress response. In addition we have been using this protocol for 10 years to assess the reproducibility of our model of depression-like behavior (and it is in all our publication on this model).

Gregus A, Wintink AJ, Davis AC, Kalynchuk LE. Effect of repeated corticosterone injections and restraint stress on anxiety and depression-like behavior in male rats. Behav Brain Res. 2005 Jan 6;156(1):105-14. doi: 10.1016/j.bbr.2004.05.013. PMID: 15474655.

Antonio Armario. The forced swim test: Historical, conceptual and methodological considerations and its relationship with individual behavioral traits. Neuroscience & Biobehavioral Reviews Volume 128, September 2021, Pages 74-86

  • Details should be provided regarding the antidepressant administration: dose, volume of injection, timing of injection before the behavioural tests, origin of purchase…
  • Details should be provided regarding the possible anesthetic method used during the sacrifice (decapitation) of the animals.

We are grateful to Reviewer 1 for pointing out the missing information. We have indicated this information in the manuscript (Page 17, line 540-545; Page 18, line 586-588). Clomipramine (Sigma-Aldrich) was administered daily at 10 am intranasally or intraperitoneally at a dose of 7.5 mg/kg for the duration of ultrasound exposure from days 1 to 21. IP Clomipramine was administered in volume of 500 μL, IN - 30 μL. Each rat was gently held with the ventral side up and drug was pipetted with 5 μL delivered to each nostril (total three times in each nostril). Once all the behavioral tests were completed the rats were anaesthetized with isoflurane (4%, 2 L/min) in a chamber until disappearing the toe reflex. Then the rats were quickly sacrificed by decapitation.

  • The authors should be consistent in using ”frontal cortex“ or “prefrontal cortex” in the whole manuscript, as the prefrontal cortex is only a part of the frontal cortex. By the way, how did the authors harvest precisely the frontal cortex/prefrontal cortex and hippocampus of the rats? It is important to precise the anatomical boundaries of these two brain regions.

We dissected the frontal cortex and indicated it throughout the manuscript. Hippocampus and frontal cortex were isolated as described by Chiu et al., 2007.

Chiu K, Lau WM, Lau HT, So KF, Chang RCC (2007) Micro‑dissection of rat brain for RNA or protein extraction from specific brain region. J Vis Exp. 7: 269.

  • Avoid abbreviations in the titles of the figures/tables and define abbreviations used in the legend.

Correction was performed.

  • Table 3 (pathway analysis) may be put into the supplement. It contains redundant data with Figure 4.

Following a recommendation of Reviewer 1, we put the table in supplement.

  • Why did the authors not characterize the hypothalamic–pituitary–adrenal axis activity and reactivity? Alterations in the hypothalamic–pituitary–adrenal axis is considered an endophenotype of depression and found in many animal models of depression. 

We thank reviewer 1 for this remark, we agree that the hypothalamic-pituitary-adrenal axis has a major contribution to the pathophysiology of depression. We evaluated  levels of corticosterone, epinephrine and norepinephrine in blood in our previous article, which was mentioned in the manuscript. But we added sentence about it in introduction: Alterations in the hypothalamic-pituitary-adrenal axis are observed in depression and shown also in animal models. In the US model of depression, the level of epinephrine, norepinephrine and corticosterone was elevated after US stress. (Page 2, line 91-95)

 Zorkina YA, Zubkov EA, Morozova AY, Ushakova VM, Chekhonin VP. The Comparison of a New Ultrasound-Induced Depression Model to the Chronic Mild Stress Paradigm. Front Behav Neurosci. 2019 Jul 2;13:146. doi: 10.3389/fnbeh.2019.00146. PMID: 31312126; PMCID: PMC6614435.

  • Why did the authors not inject the vehicle to the US rats, to be comparable with the US rats receiving Clom IP or IN??

We apologize for forgetting to add information about details about drugs in materials and methods. We are grateful to Reviewer 1 for bringing this to our attention. We added this sentence in material and method section:  US and Control group received intranasal saline in volume of 30 μL for 21 day. (Page 17, line 544-545)

  • The numbering of the figures and tables should be checked (there is 2 Tbales 3) and the referencing in the main text to the tables and figures should be adjusted accordingly. 

Correction was performed.

  • Please check how to report the outcomes of the ANOVA test and the pairwise comparison in the whole Result section!

Correction was performed.

  • The results of the behavioural tests are very poorly written. 
    • It would be better to describe test by test and write a short conclusion of the findings, yet without interpretation.

We are thankful to Reviewer 1 for this question. We have added more detail to the Result section.

  • Why represent behavioural data as box plots (which should be defined) when conducting parametric analyses? 

We replaced box plots with a histogram (median±SD), but the Morris test has a non-normal distribution.

  • Please choose different symbols for the pairwise comparisons: ultrasound versus control rats, clomipramine versus ultrasound rats and ultrasound versus control rats. Please check that the symbols are positioned correctly above the graph bars.
    • The symbol # above the IP Clom in the sucrose preference test does not look at the right place as it is unlikely that the US and the IP Clom significantly differed in the sucrose index (visually speaking).

We are grateful to Reviewer 1 for this remark. We presented the information more clearly.

  • Homogenize the use of Porsolt or Forced Swim test.

Correction was performed.

  • It would be good to separate Table 1 into 2 tables, one for the prefrontal cortex and one for the hippocampus as Table 1 is very big. Also, it would be good to use the same symbols as in the Figure 1 to highlight the pairwise comparisons (the p value does not indicate to the reader which groups are significantly different).

Reviewer 1's remark is interesting, and we thought about it, but our group of authors decided not to split the table into two parts. We want readers to compare the differences in concentration between hippocampus and frontal cortex.

The p-value in this table reflects the comparison of all groups before post hoc analysis. The reader can see the results of the post hoc analysis in supplementary Figures S4 and S6. We have added this sentence to the table description: The results of the post hoc analysis are in supplementary (Figures S4 and S6).

  • Not enough details are provided for Figure 2.
    • In Figure 2 and 3, why using Hippocamp as an abbreviation for hippocampus??

Correction was performed.

  • The Table 2 contains statistical outputs and should be in the supplement.

Correction was performed.

  • Line 182: it cannot be postdoc ANOVA, please correct.

For sure it can be, nevertheless we replaced it with comparison of regression slopes

  • The heat-maps Figure 3 is the representation of Table 1, somehow I do not catch the differences and how the normalization of the data from Table 1 to Figure 3 was performed.

  • A better explanation of the choice of the metabolic pathways by brain region from the results in Table 1 should be provided.

We based our choice of metabolic pathways on Pathways analysis in the MetaboAnalyst 5.0 service, which is described in sections 2.5 and 4.6

  • The justification of the method used lines 211-213 should be put in the Method section.

Correction was performed.

  • The Discussion section is very lengthy and definitely lacks a good structure. It should be rewritten to highlight the aims and results of the study, and then provide some interpretation of the data as well as limitations and perspectives. The Discussion provided by the authors is only reflecting, to my opinion, text-book knowledge about the different metabolic pathways altered in the rat model of depression.
    • The Discussion section should be entirely rewritten to summarize the results and interpret them regarding the existing literature and possible implications for our understanding of major depression.

We are thankful to Reviewer 1 for this question. We rewrote the Discussion section in accordance with Reviewer 1's remarks, and it has greatly improved this part of the manuscript.

  • The use of the forced swim test should be referred to as stress-coping and not “depression”-like behaviours, please see papers PMID: 28287253 and PMID: 30738104.

We are grateful to Reviewer 1 for the interesting and educational reviews. However, according to Molendijk (2021), 60% of researchers tend to label the immobility in forced swim test as depression-like behavior. We perform the forced swim test in conjunction with other tests for depression, so considering all the tests together, including forced swim, we can consider it as depression-like state. However, we removed the phrase "depression-like behavior" where it described the forced swim test separately in the Results section.

Molendijk ML, de Kloet ER. Forced swim stressor: Trends in usage and mechanistic consideration. Eur J Neurosci. 2021 Feb 6. doi: 10.1111/ejn.15139. Epub ahead of print. PMID: 33548153.

  • The alterations in mitochondria in depression have been recently reviewed in PMID: 32979495.

We found this article to be informative, although we removed the mention of mitochondria in the process of remaking the discussion.

  • Please make sure to define all abbreviations before first mention and avoid starting a sentence with an abbreviation or a number.

Correction was performed.

  • Make sure to homogeneize the past tense in the Method and Result sections.

Correction was performed.

Reviewer 2 Report

The manuscript entitled "Brain metabolic profile after intranasal vs. intraperitoneal clomipramine treatment in rats with ultrasound model of depression" by Abramova et al. provides metabolomic data for prefrontal cortex and hippocampus of rats with depression-like behaviour induced with ultrasound. To the best of my knowledge it is the first study describing changes in metabolites level in those two structures in ultrasound-induced depressive animals, what makes the study very interesting. However, I have few concerns which I divided into major and minor:

Major concerns:

The manuscript is very laconic regarding one of the most important thing for the study - brain samples collection. I would like the authors to describe how the animals were killed. Were the animals anaesthetized and decapitated, or after the anaesthetics administration the chest was opened and the animal was perfunded with physiological solution throught the heart? It is very vital for the study because even a very small amount of blood remaining within the cerebral vessels could interfered with the chromatographic analysis.

There is no information about the clomipramine administration. What was the concentration of the drug used for intranasal and intraperitoneal treatment? When precisely the drug was administrated? Was it a single dose or the drug was administrated repeatedly?

Do the authors used the chromatographic analysis to establish the concentration of clomipramine itself in the brain samples? Were there any differences between intranasal and intraperitoneal administration?

Minor concerns:

Please improve the visibility of significance in figure 1.

The authors should re-read the manuscript carefuly and decide if the analyzed brain part was frontal cortex or prefrontal cortex. It is not the same.

Please correct the decimal separator in table 1 from comma into dot.

What were the sex and age of the animals used in the study?

Please correct the bibliography according to the guide for authors.

Author Response

We would like to thank Reviewer 2 for favorable comments to our work and valuable suggestions to improve the manuscript. We added some information according to his recommendations and corrected our manuscript.

We presented our paper corrections in more understandable way according to yours recommendations. For cross-correction procedure simplicity we numbered lines and pages in the text and specified corresponding numbers for commentary to questions. We have highlighted in yellow the corrections in the text. We hope that such form of answer will be appropriate.

Answer on questions from Reviewer 2: 

  • The manuscript is very laconic regarding one of the most important thing for the study - brain samples collection. I would like the authors to describe how the animals were killed. Were the animals anaesthetized and decapitated, or after the anaesthetics administration the chest was opened and the animal was perfunded with physiological solution throught the heart? It is very vital for the study because even a very small amount of blood remaining within the cerebral vessels could interfered with the chromatographic analysis.

We are grateful to Reviewer 2 for the important observation. We did not perfuse with saline because perfusion requires a longer anesthesia time and brings a lot of stress and pain to the animal, we are concerned that it might affect the results of the brain tissue analysis. In addition, some evidence suggests that perfusion has no effect when using the chromatographic method (Fenyk-Melody, 2004). Although the type of anesthesia can add variation to the metabolic profile, it is the timing of organ harvesting and freezing that most affects the metabolome (Maksimovic, 2019).

We apologize for forgetting to add information about details about drugs in materials and methods. We are grateful to Reviewer 2 for bringing this to our attention. We added this sentence in material and method section: Once all the behavioral tests were completed the rats were anaesthetized with isoflurane (4%, 2 L/min) in a chamber until disappearing the toe reflex. Then the rats were quickly sacrificed by decapitation. (Page 18, line 586-588)

Fenyk-Melody JE, Shen X, Peng Q, Pikounis W, Colwell L, Pivnichny J, Anderson LC, Tamvakopoulos CS. Comparison of the effects of perfusion in determining brain penetration (brain-to-plasma ratios) of small molecules in rats. Comp Med. 2004 Aug;54(4):378-81. PMID: 15357317.

Maksimovic I, Zhang S, Vuckovic I, Irazabal MV. Kidney harvesting and metabolite extraction for metabolomics studies in rodents. Methods Cell Biol. 2019;154:15-29. doi: 10.1016/bs.mcb.2019.05.009. Epub 2019 Jul 26. PMID: 31493816; PMCID: PMC6865266.

  • There is no information about the clomipramine administration. What was the concentration of the drug used for intranasal and intraperitoneal treatment? When precisely the drug was administrated? Was it a single dose or the drug was administrated repeatedly?

We are grateful to Reviewer 2 for bringing this to our attention. We added this sentence in material and method section:  US and Control group received intranasal saline in volume of 30 μL for 21 day. (Page 17, line 544-545)

  • Do the authors used the chromatographic analysis to establish the concentration of clomipramine itself in the brain samples? Were there any differences between intranasal and intraperitoneal administration?

Intranasal administration is used, among other things, to increase permeability through the hemato-encephalic barrier and increase the bioavailability of drugs. We have not determined clomipramine concentrations, but may do so in other experiments.

Please improve the visibility of significance in figure 1.

We have changed the visibility of significance in Figure 1.

The authors should re-read the manuscript carefully and decide if the analyzed brain part was frontal cortex or prefrontal cortex. It is not the same.

We are grateful for this remark, this errors is fixed now. We dissected the frontal cortex and indicated it throughout the manuscript.

Please correct the decimal separator in table 1 from comma into dot.

Correction was performed.

What were the sex and age of the animals used in the study?

We added information to Materials and Methods on the sex and age of animals (Page 16, line 515-520).

Please correct the bibliography according to the guide for authors.

Correction was performed.

Round 2

Reviewer 1 Report

I am satisfied with the revisions made on the manuscript. The authors have addressed all the points raised and answered all comments/concerns. The scientific writing, the structure, the graphical representations and the discussion have been greatly approved. The manuscript is, to my opinion, suitable for publication. I think that it brings interesting and new insights about the possible alterations in metabolism involved in depression, and I am looking forward to further studies using the animal model of depression.

Reviewer 2 Report

Thank you very much for addressing all of my concerns.